# A Systematic Study of Gate Functions in Soft Adaptive Policy Optimization

**Egor Denisov[2], Svetlana Glazyrina[2], Maksim Kryzhanovskiy[2], Roman Ischenko[1,2]**

[1]Institute for Artificial Intelligence, Lomonosov Moscow State University, Moscow, Russia
[2]Lomonosov Moscow State University, Moscow, Russia

## Abstract

Group Relative Policy Optimization (GRPO) has significantly advanced the training of large language models and enhanced their reasoning capabilities, while it remains susceptible to instability due to the use of hard clipping. Soft Adaptive Policy Optimization (SAPO) addresses this limitation by replacing clipping with a smooth sigmoid-based gate function, which leads to more stable updates. We push this theory further and investigate the impact of different gate functions on both training stability and final model performance. We formalize the key properties that admissible gates should satisfy and propose several families of such functions for empirical evaluation. This paper presents an analysis of our findings based on experiments conducted with the Qwen2.5-7B-Instruct model on mathematical reasoning tasks. These results provide practical guidance for designing smoother and more robust policy optimization objectives for large language model training.

## 1 Introduction

Reinforcement learning (RL) has become a central tool for training large language models, enabling them to solve some of the most challenging problems across a wide range of different domains (Qwen-Team, 2025; DeepSeek-AI, 2025; Gemini-Team, 2025). The rapid progress in this field, together with the continuous growth in the scale of training data, necessitates the development of training algorithms that are both stable and highly scalable.

One of the most widely used RL algorithms in current practice is Group Relative Policy Optimization (GRPO) (Shao et al., 2024; DeepSeek-AI, 2026). GRPO can be considered as a development of Proximal Policy Optimization (PPO) (Schulman et al., 2017a) that eliminates the need for a separate value function network for advantage estimation. Instead, GRPO samples multiple outputs for the same query and computes group-based advantage estimates, leading to a reduction in computational demand and training cost. Similar to PPO, GRPO supports off-policy updates through the use of importance sampling, enabling the reuse of trajectories collected under previous policies and thereby improving sample efficiency. However, large deviations of the importance sampling ratio from unity may result in unstable training dynamics. To mitigate this issue, GRPO adopts the PPO clipping mechanism, which constrains policy updates and maintains proximity to the reference policy.

While clipping is practically effective, it cannot be regarded as a universally applicable method. Its hyperparameters are difficult to tune in order to balance training stability and exploration. A wide clipping range allows large importance ratio deviations to produce noisy and potentially destabilizing gradients, whereas a narrow clipping range suppresses most gradients, resulting in minimal policy updates, lack of exploration, and stagnated learning.

This limitation is addressed by Soft Adaptive Policy Optimization (SAPO) (Gao et al., 2025), which replaces hard clipping with a continuously differentiable gate function. The gradient of this function attains its maximum at an importance ratio of one and smoothly decays as the ratio moves away from this point. This design enables a gradual suppression of updates corresponding to extreme deviations from the reference policy, while preserving non-zero gradients and on-policy behavior. Additionally, SAPO introduces a temperature parameter that depends on the sign of the advantage, allowing the algorithm to more strongly encourage beneficial policy updates while being more conservative when penalizing unfavorable ones.

The behavior of gate-based methods is largely determined by the effective width of the gradient peak around its maximum, as well as by the decay characteristics of the gradient for larger deviations of the argument. In the original SAPO paper, a sigmoid function is used, whose gradient decays exponentially. In this work, we explore alternative families of soft gates exhibiting diverse gradient decay behaviors, ranging from polynomial to Gaussian attenuation.

Our contributions are:

- We formalize the key properties that gate functions should satisfy and introduce several novel families of such functions.
- We propose a new metric, *Effective Update Ratio (EUR)*, which generalizes the *clip ratio* used in GRPO and provides a unified view of token-level policy updates.
- We conduct a comprehensive empirical study to assess how the choice of gate function affects training dynamics and downstream performance, evaluating our approach on mathematical reasoning benchmarks.

## 2   RELATED WORKS

Reinforcement learning has become a central paradigm for fine-tuning large language models beyond supervised approaches. Policy gradient methods typically rely on importance sampling to correct for distribution mismatch between the behavior and target policies. Trust Region Policy Optimization (TRPO) (Schulman et al., 2017b) enforces an explicit constraint on the Kullback–Leibler divergence between successive policies, while Proximal Policy Optimization (PPO) (Schulman et al., 2017a) addresses instability by introducing a clipped surrogate objective that restricts policy updates.

GRPO (Shao et al., 2024; DeepSeek-AI, 2026) was proposed as a scalable and value-free alternative to PPO, which replaces value function estimation with group-wise relative advantage normalization computed over multiple sampled outputs for the same prompt. Several extensions and variants of GRPO have been proposed to further improve stability and sample efficiency. GSPO (Zheng, 2025) implements sequence-level importance weights; GMPO (Zhao, 2025) modifies the aggregation of group statistics to reduce sensitivity to outliers; Dr. GRPO (Liu, 2025) removes normalization to avoid optimization bias; DAPO (Seed, 2025) applies dynamic sampling and decouples clipping bounds. Nevertheless, most GRPO-style methods retain PPO-inspired hard clipping and may inherit unstable updates and entropy collapse.

Several studies investigate alternatives to hard clipping with the aim of improving robustness (Wang et al., 2025; Sun et al., 2022; Su et al., 2025; Han et al., 2019; Garg et al., 2021; MiniMax, 2025). Soft Adaptive Policy Optimization (SAPO) (Gao et al., 2025), inspired by (Chen et al., 2022), leverages a smooth gate function that progressively down-weights samples with large importance ratios.

Despite the success of SAPO, there remains significant room for investigating the impact of an appropriate gate function on the method's performance.

## 3   PRELIMINARIES

**Group Relative Policy Optimization (GRPO).**   Let $\mathcal{Q}$ denote a query set ($\mathcal{Q} = \{q_i\}_{i=1}^{|\mathcal{Q}|}$). In GRPO (Shao et al., 2024; DeepSeek-AI, 2026) for each query $q \in \mathcal{Q}$, a group of $G$ responses $\{o_1, \cdots, o_G\}$ from the behavior policy $\pi_{\theta_{old}}$ is sampled. The policy is being optimized by maximizing the following objective:

$$\mathcal{J}_{\text{GRPO}}(\theta) = \mathbb{E}_{q \sim \mathcal{Q}, \{o_i\}_{i=1}^{G} \sim \pi_{\theta_{old}}(.|q)}$$

$$\left[ \frac{1}{G} \sum_{i=1}^{G} \frac{1}{|o_i|} \sum_{t=1}^{|o_i|} \min\left( r_{i,t}(\theta) \hat{A}_{i,t}, \text{clip}(r_{i,t}(\theta), 1 - \varepsilon, 1 + \varepsilon) \hat{A}_{i,t} \right) \right] \quad (1)$$

where $\{R_1, \cdots, R_G\}$ are rewards for each response, $r_{i,t}(\theta) = \frac{\pi_\theta(o_{i,t}|q, o_{i,<t})}{\pi_{\theta_{old}}(o_{i,t}|q, o_{i,<t})}$ is the importance sampling ratio, $\hat{A}_{i,t} = \frac{R_i - \text{mean}(R)}{\text{std}(R)}$ is the normalized advantage of the $i$-th response, $\varepsilon > 0$ is the clipping threshold.

**Soft Adaptive Policy Optimization (SAPO).** SAPO (Gao et al., 2025) generalizes the idea of GRPO by introducing a smooth gate function instead of hard clipping. The objective transforms into the following:

$$\mathcal{J}_{SAPO}(\theta) = \mathbb{E}_{q \sim \mathcal{Q}, \{o_i\}_{i=1}^G \sim \pi_{\theta_{old}}(.|q)} \left[ \frac{1}{G} \sum_{i=1}^G \frac{1}{|o_i|} \sum_{t=1}^{|o_i|} f_{i,t}(r_{i,t}(\theta)) \hat{A}_{i,t} \right], \tag{2}$$

where

$$f_{i,t}(x) = \sigma(\tau_{i,t}(x-1)) \cdot \frac{4}{\tau_{i,t}}, \quad \sigma(x) = \frac{1}{1+e^{-x}}, \quad \tau_{i,t} = \begin{cases} \tau_{pos}, & \hat{A}_{i,t} > 0 \\ \tau_{neg}, & \text{otherwise} \end{cases} \tag{3}$$

Gradient of the objective becomes:

$$\nabla_\theta \mathcal{J}_{SAPO}(\theta) = \mathbb{E}_{q \sim \mathcal{Q}, \{o_i\}_{i=1}^G \sim \pi_{\theta_{old}}(.|q)}$$
$$\left[ \frac{1}{G} \sum_{i=1}^G \frac{1}{|o_i|} \sum_{t=1}^{|o_i|} w_{i,t}(\theta) r_{i,t}(\theta) \nabla_\theta \log \pi_\theta(o_{i,t} \mid q, o_{i,<t}) \hat{A}_{i,t} \right], \tag{4}$$

where

$$w_{i,t}(\theta) = \left. \frac{\mathrm{d} f_{i,t}}{\mathrm{d} x} \right|_{x=r_{i,t}(\theta)} = \tau f_{i,t}(r_{i,t}(\theta))(1 - \frac{\tau}{4} f_{i,t}(r_{i,t}(\theta))) \tag{5}$$

$w_{i,t}(\theta)$ reaches its maximum equal to 1 at $r_{i,t}(\theta) = 1$, which corresponds to the on-policy behavior. As the importance ratio deviates from 1, the gradient starts decreasing exponentially towards 0. This allows more stable training without limiting model exploration.

## 4 METHODOLOGY

We hypothesize that an appropriate choice of the gate function in the SAPO algorithm can lead to improved convergence during model training by enabling an optimal contribution of individual tokens in the optimization process. Before exploring and analyzing alternatives, it is necessary to formalize a set of properties that all candidate functions $f : \mathbb{R}_+ \to \mathbb{R}$ should satisfy: (i) the function should be continuously differentiable; (ii) the derivative $f'(x)$ should attain its global maximum that equals to 1 at $x = 1$; (iii) the derivative should decrease monotonically as the argument moves away from 1; and (iv) $f'(x) \cdot x \to 0$ as $x \to \infty$.

The suggested properties admit a clear optimization interpretation. (ii) ensures that when $r_{i,t}(\theta) = 1$, the resulting gradient contribution coincides with the on-policy update, thereby preserving consistency with standard policy gradient optimization. (iii) enforces a controlled attenuation of samples whose importance ratios deviate from 1. This behavior stabilizes training by discouraging overly aggressive updates induced by off-policy samples. Finally, (iv) guarantees that samples associated with extreme importance ratios contribute negligibly to the gradient, effectively suppressing the impact of outliers and preventing instability caused by heavy-tailed ratio distributions.

We consider the sigmoid applied in the original paper as a baseline and propose the following alternatives for the gate function: Error function (Normal CDF) (6), Arctangent (7) and Softsign (8). Consistent with the original paper, we incorporate a temperature parameter $\tau$ into the objective. The additive constants introduced in the gate functions do not affect their gradients and are included solely for interpretability. Specifically, these constants ensure that the resulting gate functions remain positive on the half-interval $[0; +\infty)$ and are normalized to pass through the point $(1, 1)$.

$$f_{\mathrm{erf}}(x) = \sqrt{\frac{\pi}{2\tau^2}} \left( 1 + \mathrm{erf}\left( \frac{\tau(x-1)}{\sqrt{2}} \right) \right) + 1 - \sqrt{\frac{\pi}{2\tau^2}}, \quad f'_{\mathrm{erf}}(x) = \exp\left( -\frac{\tau^2(x-1)^2}{2} \right) \tag{6}$$

$$f_{\mathrm{arctan}}(x) = 1 + \frac{1}{\tau} \arctan(\tau(x-1)), \quad f'_{\mathrm{arctan}}(x) = \frac{1}{(1+\tau^2(x-1)^2)} \tag{7}$$

$$f_{\mathrm{softsign}}(x) = 1 + \frac{x-1}{\sqrt{1+\tau^2(x-1)^2}}, \quad f'_{\mathrm{softsign}}(x) = \frac{1}{(1+\tau^2(x-1)^2)^{3/2}} \tag{8}$$

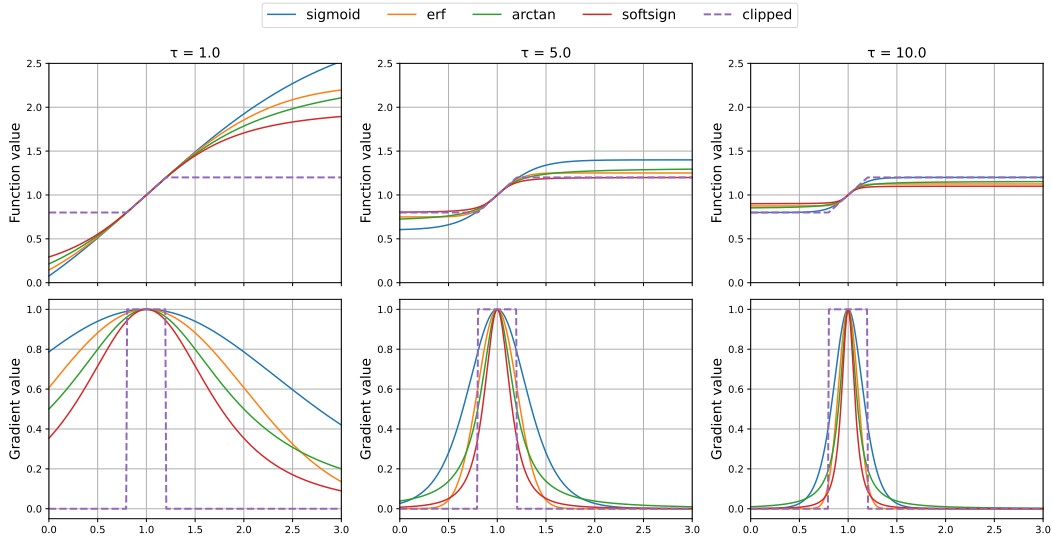

Figure 1: Temperature-dependent behavior of the considered gate functions (top row) and their gradients (bottom row) for $\tau \in \{1, 5, 10\}$. All functions are normalized to be positive on $[0; +\infty)$ and to pass through the point $(1, 1)$. Increasing the temperature sharpens the transition around $x = 1$, leading to more localized and higher gradient peaks for smooth gates, while the clipped variant exhibits piecewise-linear behavior with constant gradients in its active region.

The behavior of the selected functions, in comparison with hard clipping and the sigmoid function used in the original SAPO formulation, is illustrated in Figure 1.

While all considered functions satisfy the aforementioned properties, they differ substantially in the behavior of their derivatives. The derivatives of arctangent and softsign exhibit polynomial decay, and the derivative of the error function follows a Gaussian decay. Heavier-tailed gradients amplify the influence of rare tokens on the learning process, thereby increasing exploration, albeit potentially at the cost of reduced stability. Conversely, lighter-tailed gradients of $f_{\mathrm{erf}}$ make the method increasingly similar to hard clipping by progressively suppressing the contribution of extreme tokens.

## 5 EXPERIMENTS

### 5.1 EXPERIMENTAL SETUP

All experiments were conducted using the Qwen2.5-7B-Instruct model on mathematical reasoning tasks.

**Training.** The training data consisted of an equal mixture of the GSM8K (train split) (Cobbe et al., 2021) and DeepMath(He et al., 2025) datasets, and the model was aligned via a cold-start reinforcement learning procedure without supervised warm-up. For each prompt, we generated 8 rollouts with a maximum response length of 512 tokens. The per-device batch size was set to 1, and gradients were accumulated over 16 steps, yielding a larger effective batch size. To improve sample efficiency and to better isolate the effect of gate function alteration, two gradient updates were performed for each batch, enabling reuse of previously generated trajectories. Training was performed on 8 NVIDIA A100 GPUs for a total of 5,000 optimization steps. Generation parameters are provided at Appendix A.

**Reward.** The reward function was defined as a sum of an answer correctness component and a formatting component:

$$r = r_{\mathrm{answer}} + r_{\mathrm{format}}. \tag{9}$$

$r_{format}$ evaluates whether the model follows a prescribed structured response template of the form *<think> text </think> <answer> text </answer>*, where *<think>, </think>, <answer>* and

*</answer>* denote special tokens and *text* represents arbitrary generated content:

$$r_{\text{format}} = \begin{cases} 1, & \text{format is fully satisfied} \\ 0.5, & \text{generation begins with the } \textit{<think>} \text{ and ends with the } \textit{</answer>} \\ 0.25, & \text{generation begins with the } \textit{<think>} \text{ or ends with the } \textit{</answer>} \\ 0, & \text{otherwise} \end{cases} . \quad (10)$$

$r_{answer}$ measures solution correctness. We extract the text generated by the model between the *<answer>* and *</answer>* tokens and evaluate it for consistency with the ground-truth solution.

$$r_{\text{answer}} = \begin{cases} 1, & \text{model output matches the correct answer} \\ 0, & \text{otherwise} \end{cases} . \quad (11)$$

**Evaluation.** The trained models were evaluated on several mathematical benchmarks of varying difficulty: GSM8K (test split), MATH500 and AIME. GSM8K (Cobbe et al., 2021) consists of grade school arithmetic word problems requiring multi-step reasoning. MATH500 is a curated subset of the MATH (Hendrycks et al., 2021) dataset containing competition-level problems across algebra, geometry, number theory, and combinatorics. AIME comprises problems from the corresponding editions of the American Invitational Mathematics Examination, designed to assess advanced mathematical reasoning and precise numerical answer generation. During evaluation, responses were generated with a sampling temperature of $0.7$. We used the accuracy metric for measuring model performance.

## 5.2 RESULTS

**Rewards.** We explore several temperature settings for the proposed gate functions. As shown in Figure 2, for the erf-based gate temperature hyperparameter pair $\tau_{pos} = 10, \tau_{neg} = 12$ yields the best results. These values provide an optimal trade-off between aggressive clipping ($\tau_{pos} = 13, \tau_{neg} = 15$) and fully accounting for the contribution of all tokens ($\tau_{pos} = 1, \tau_{neg} = 1.05$). Similar behavior was observed across all considered gate functions including sigmoid from SAPO. Therefore, in the subsequent comparative analysis, we report results corresponding to this temperature configuration.

Figure 3 illustrates reward dynamics for the considered methods in comparison with original SAPO and GRPO with $\varepsilon = 0.2$. At the early stages of training, all SAPO-based variants demonstrate steeper increase in rewards. Leveraging soft gates leads to faster exploration and facilitates 4 times quicker acquisition of the correct response format. Overall, the use of the error function and the arctangent gating mechanisms results in superior performance.

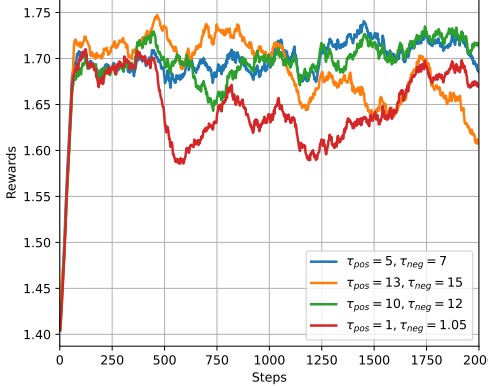
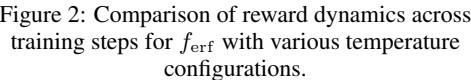
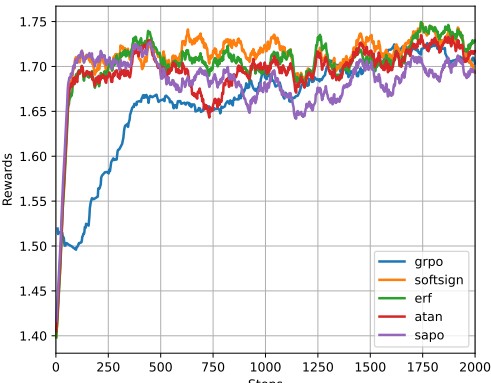

Figure 2: Comparison of reward dynamics across training steps for $f_{\text{erf}}$ with various temperature configurations.

Figure 3: Comparison of reward dynamics across training steps for all considered methods with best configurations. For GRPO $\varepsilon = 0.2$, for SAPO-like methods $\tau_{pos} = 10, \tau_{neg} = 12$.

**Entropy and EUR.** In addition, we investigate the behavior of policy entropy depending on both the chosen method and the temperature parameter. We introduce a quantitative indicator termed

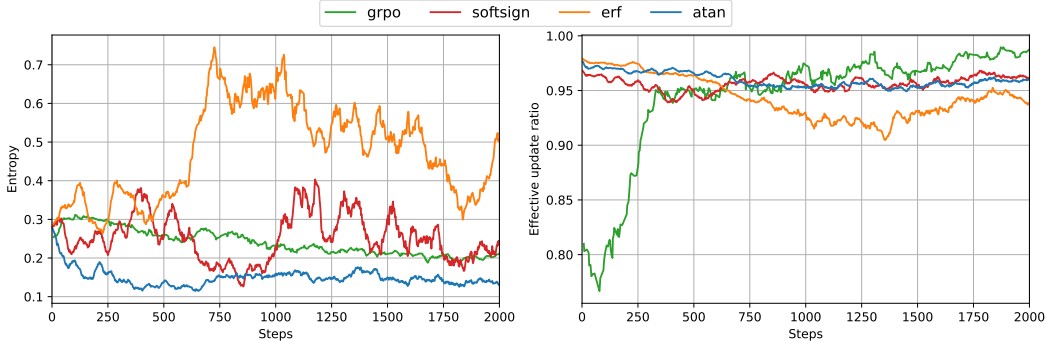

Figure 4: Policy entropy (left) and effective update ratio (right) over training for different gate functions and GRPO. For all SAPO-like methods $\tau_{pos}$ is set to 10 and $\tau_{neg}$ is set to 12.

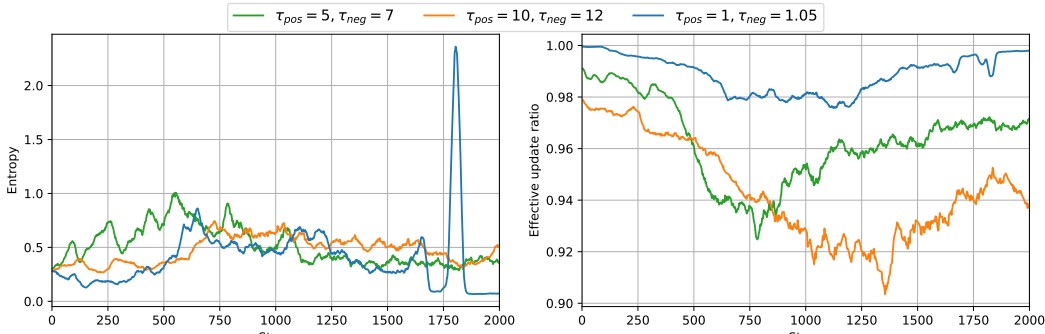

Figure 5: Policy entropy (left) and effective update ratio (right) over training for $f_{\mathrm{erf}}$ with different temperature configurations.

*Effective Update Ratio* (EUR). EUR is defined as the derivative of the gate function evaluated at $r_{i,t}(\theta)$ averaged over all tokens used at the current training step:

$$\mathrm{EUR} = \frac{1}{B} \sum_{i=1}^{B} \frac{1}{\sum_{j=1}^{G} |o_j|} \sum_{j=1}^{G} \sum_{k=1}^{|o_j|} f'_{j,k}(r_{j,k}(\theta)) \tag{12}$$

where B represents batch size, G is the number of outputs for each prompt. For GRPO we have $f'_{j,k}(r_{j,k}(\theta)) = \begin{cases} \mathbf{1}[r_{j,k}(\theta) \le 1 + \varepsilon], & \hat{A}_j > 0 \\ \mathbf{1}[r_{j,k}(\theta) \ge 1 - \varepsilon], & \hat{A}_j \le 0 \end{cases}$ , where $\mathbf{1}[.]$ denotes the indicator function. Under this definition, the EUR for GRPO reduces to $1 - clip\_ratio$, where *clip_ratio* is a fraction of tokens with importance sampling ratio out of clipping range. EUR takes values from $[0, 1]$ and can be interpreted as the mean fraction of token contribution to the parameter update at a given training step.

As observed from Figure 4, an increase in policy entropy is associated with a decrease in EUR. This effect can be explained by the higher probability of sampling tokens with large importance ratios that significantly deviate from 1. Such tokens contribute less to the policy update and, consequently, reduce the Effective Update Ratio. Thus gates still act as a form of a soft policy constraint to prevent destabilizing updates and the following memory loss.

Figure 5 illustrates the dependence of policy entropy and EUR for $f_{\mathrm{erf}}$ on the temperature parameter. We discover that increasing the temperature leads to a decrease in EUR, as the gradient becomes more sharply peaked around 1 and decays more rapidly away from this point. At lower temperatures ($\tau = 5$), the entropy reaches higher values (for $\tau = 1$ training becomes unstable and collapses). Nevertheless, even a high value of temperature parameter is sufficient to prevent the entropy decay observed in GRPO.

**Benchmark results.** Table 1 summarizes results of best configurations for each gate function to assess their empirical performance on mathematical tasks. The best results on each benchmark are highlighted in bold.

| Model | GSM8K | MATH500 | AIME |
|---|---|---|---|
| Qwen2.5-7B-Instruct | 80.3 | 56.6 | 3.3 |
| + GRPO | 80.6 | **58.6** | 8.3 |
| + SAPO (sigmoid baseline) | 82.2 | 57.8 | 8.3 |
| + SAPO (erf) | 82.7 | 56.2 | **10.0** |
| + SAPO (arctan) | 82.3 | 55.8 | 5.0 |
| + SAPO (softsign) | **82.8** | 57.2 | 5.0 |

Table 1: Accuracy (%) of different model configurations on mathematical reasoning benchmarks.

The methods we proposed achieve improvements on most benchmarks (82.8% for $f_{\text{softsign}}$ on GSM8K and 10% for $f_{\text{erf}}$ on AIME). To fully realize their potential, a more careful selection of training data could be considered, which would enable the model to perform more extensive reasoning.

## 6 CONCLUSION

We formalized the key properties that gate functions must satisfy to ensure stable and effective policy updates. We introduced several novel gate function variants for SAPO and systematically investigated their impact on the method's behavior. In addition, we analyzed these approaches through the lens of entropy dynamics and a newly proposed metric, the Effective Update Ratio, which quantifies the relative contribution of tokens to the policy update at each training step. Our empirical results indicate that the erf-based gate shows the strongest gains on AIME and improves GSM8K vs. sigmoid baseline, while MATH500 remains challenging. Furthermore, we observe that overall performance is highly sensitive to the choice of temperature parameters, which must strike a careful balance between overly aggressive clipping and excessively smooth updates that incorporate nearly all tokens. This sensitivity suggests that principled temperature selection and adaptive scheduling strategies constitute a promising direction for future research.

## COMPUTATIONAL RESOURCES

The research was carried out using the MSU-270 supercomputer of Lomonosov Moscow State University.

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

# APPENDICES

## A TRAINING PARAMETERS

Table 2 summarizes the training and generation hyperparameters used in our experiments. All configurations were kept fixed across runs unless explicitly stated otherwise.

| Category | Parameter | Value |
|---|---|---|
| Generation | MAX_PROMPT_LENGTH | 1536 |
| | MAX_COMPLETION_LENGTH | 512 |
| | NUM_GENERATIONS | 8 |
| Batching | PER_DEVICE_TRAIN_BATCH_SIZE | 1 |
| | GRAD_ACCUM | 16 |
| | NUM_ITERATIONS | 2 |
| Precision | BF_16 | TRUE |
| Optimization | OPTIMIZER | AdamW |
| | LEARNING_RATE | $10^{-6}$ |
| | WARMUP_STEPS | 0 |
| | SCHEDULER_TYPE | linear |
| AdamW Parameters | ADAM_BETA1 | 0.9 |
| | ADAM_BETA2 | 0.999 |
| | ADAM_EPSILON | $10^{-8}$ |

Table 2: Training and generation hyperparameters used across all experiments.

