# OpenReview forum: "A Systematic Study of Gate Functions in Soft Adaptive Policy Optimization"
_mathai.club/MathAI/2026/Conference — 2026 Oral_

### Official Review · Reviewer_8T6j · 2026-03-11
**A well-structured empirical analysis of gate functions for stabilizing SAPO/GRPO training in LLM reinforcement learning, providing practical insights though with limited novelty and evaluation scope.**

**Rating:** 7
**Confidence:** 4

**Review:**

Summary

This paper studies gate functions used in Soft Adaptive Policy Optimization (SAPO) as an alternative to hard clipping in Group Relative Policy Optimization (GRPO). The authors formalize desirable mathematical properties of gate functions and empirically evaluate several families of smooth gating mechanisms on Qwen2.5-7B-Instruct for mathematical reasoning tasks. The goal is to improve training stability and final performance in reinforcement learning for large language models.

Strengths
1. Clear problem motivation

The instability caused by hard clipping in GRPO/PPO-style objectives is well known. Investigating smooth gate alternatives is a meaningful direction.

2. Systematic framework

The paper contributes a formal set of properties for admissible gate functions, which helps organize the design space rather than proposing a single heuristic.

3. Empirical validation on a modern LLM

Using Qwen2.5-7B-Instruct and mathematical reasoning benchmarks gives practical relevance to the analysis.

4. Practical implications

Results provide guidance for designing stable RL objectives in post-training pipelines for LLMs.

Weaknesses
1. Incremental novelty

The contribution is largely analytical and empirical rather than introducing a fundamentally new RL optimization algorithm.

2. Limited evaluation scope

Experiments appear focused on math reasoning tasks and a single model, which limits generalization to broader RLHF settings.

3. Missing comparison with broader RL variants

The study could benefit from comparisons against:

PPO variants

KL-regularized RLHF objectives

other adaptive clipping mechanisms.

4. Theoretical analysis depth

While properties of gate functions are defined, deeper theoretical guarantees (e.g., convergence or stability bounds) are limited.

---

### Official Review · Reviewer_Ltmc · 2026-03-13
**Careful and practically useful empirical study of smooth gate design for SAPO**

**Rating:** 8
**Confidence:** 4

**Review:**

This paper studies an important and timely question in RL fine-tuning for LLMs: how the choice of smooth gate function in SAPO affects training stability, exploration behavior, and downstream mathematical reasoning performance. I found the paper clear, technically coherent, and practically useful. In particular, I think the paper makes a meaningful contribution by turning what could have been a narrow heuristic tweak into a more structured empirical study: it articulates explicit admissibility properties for gate functions, evaluates several distinct gate families with different gradient-decay behavior, and introduces a simple diagnostic quantity, the Effective Update Ratio (EUR), to help interpret optimization dynamics.

The contribution is primarily empirical and practical rather than deeply theoretical, but I do think it is strong enough for acceptance. The paper addresses a real design choice in modern GRPO/SAPO-style RL pipelines, uses a relevant 7B model, and evaluates on nontrivial mathematical reasoning benchmarks. While the scope of the evidence is narrower than the broadest reading of the title might suggest, the paper still provides useful guidance for practitioners and a better conceptual organization of the gate-design space than was previously available.

**Strengths**
- The paper is well motivated. Hard clipping is simple and effective, but it is also brittle, and studying smooth alternatives is a meaningful problem for LLM RL.
- The paper gives a clean formalization of admissible gate properties instead of proposing just one more heuristic variant.
- The chosen gate families are not redundant: `erf`, `arctan`, and `softsign` correspond to meaningfully different gradient-decay regimes, so the comparison is informative.
- The experiments are practically relevant: the study is conducted on `Qwen2.5-7B-Instruct` and evaluated on `GSM8K`, `MATH500`, and `AIME`.
- The `EUR` metric is simple, but useful. It provides a more interpretable bridge between clipping-style intuitions and smooth gate behavior.
- The paper is generally well written and easy to follow.

**Main concerns**

1. **The study is systematic within a limited regime, not in the broadest possible sense.**
   The paper explores the gate-design question in a disciplined way, but the empirical scope is still fairly narrow: one base model, one overall RL recipe, one reward design, one domain family, and a relatively small set of gate families. I do not see this as a fatal issue, but it does mean the practical guidance should be interpreted as guidance for this class of math-focused SAPO training setups rather than a general statement about all RLHF or post-training settings.

2. **The theoretical contribution is useful but not deep.**
   The admissibility properties are sensible and well motivated, but the paper does not go further to provide convergence results, formal stability guarantees, or a stronger derivation of when one decay family should outperform another. In other words, the theory structures the study well, but it does not itself make the paper a major mathematical contribution.

3. **The benchmark results are mixed rather than dominated by one gate.**
   The results do show that gate choice matters, which is already a useful finding. However, the picture is not one of a single clearly superior gate: `softsign` is best on GSM8K, sigmoid SAPO is best on MATH500, and `erf` is best on AIME. I therefore think the right takeaway is not “this paper identifies the best gate,” but rather “gate shape materially affects the optimization tradeoff, and the preferred choice is benchmark- and temperature-dependent.”

4. **The empirical section would be stronger with uncertainty estimates.**
   The paper reports single accuracy numbers, but there are no multiple seeds, error bars, or significance estimates. This matters because some of the deltas are small. It matters especially for AIME, where percentage changes may correspond to only a very small number of problems. I still think the trend is interesting, but the strength of the claims should be calibrated accordingly.

5. **The reward curves are partly confounded by format reward.**
   The reward is defined as `r = r_answer + r_format`, and the format component is not negligible. This means faster reward improvement can partly reflect faster acquisition of the desired output format rather than only better reasoning or correctness. The paper is transparent about this, which I appreciate, but the interpretation of reward-based stability claims should remain somewhat cautious.

6. **The temperature-tuning protocol could be described more precisely.**
   The text presents a temperature sweep for the `erf` gate and then states that similar behavior was observed for the other functions, while Table 1 refers to the best configuration for each gate. I would like a slightly clearer explanation of how tuning fairness was ensured across all gate families.

7. **Baseline coverage could be broader.**
   GRPO and sigmoid-SAPO are the most important baselines, and including them is the right starting point. Still, the paper itself cites a broader family of modern GRPO/PPO variants, and at least one additional stronger baseline would have made the empirical story more complete.

**Clarity, originality, and significance**

On clarity, I think the paper does well. The methodology is easy to follow, and the paper succeeds in turning a fairly technical optimization-design question into something concrete and interpretable. On originality, I would characterize the contribution as incremental but meaningful: the paper does not invent an entirely new RL paradigm, but it does provide a useful synthesis and a well-designed empirical comparison that helps move beyond the default sigmoid choice in SAPO. On significance, I see this as a practically relevant contribution for RL post-training of LLMs, especially in math-oriented settings where training stability and controllable exploration matter.

**Suggestions for improvement**
- Add multi-seed evaluation or at least some uncertainty estimates for the main benchmark results.
- Clarify whether each gate family was tuned independently and with comparable effort.
- Separate the effect of format reward from answer reward more explicitly.
- Add at least one broader baseline among modern clipping or GRPO variants.
- Slightly narrow the framing of “systematic study” and “practical guidance,” unless broader evidence is added.

**Overall assessment**
I found this to be a good and useful paper. It is not a major theoretical breakthrough, and the experiments are not broad enough to justify sweeping claims about all SAPO-style optimization. But within its scope, the paper is careful, informative, and practically relevant. It gives a better understanding of how gate shape and temperature affect SAPO behavior, and it offers concrete empirical insight that I expect practitioners will find useful. For those reasons, I think it clears the bar for a clear accept.

---

### Decision · Program_Chairs · 2026-03-14

**Decision:**

Accept (Oral)

**Comment:**

Dear Author(s),

On behalf of the Program Committee of the International Conference on Mathematics of Artificial Intelligence (MathAI 2026), we are pleased to inform you that your paper has been accepted for an oral presentation at MathAI 2026.

Your paper was evaluated through a rigorous two-stage review process involving both automated screening and expert review by members of the Program Committee. The reviewers recognized the quality and contribution of your work.

Presentation details:

- Format: Oral presentation (15–20 minutes + 5 minutes Q&A)
- Mode: You may present either in person (offline) at the conference venue in Sirius, Russia, or remotely via Zoom. Please indicate your preferred mode when confirming your participation.
- Conference dates: Marh 30 - April 3, 2026
- Website: https://mathai.club

Next steps:

1. Please confirm your participation and presentation mode by replying to this email mathai.club@yandex.ru no later than March 15, 2026 18:00 Moscow time.
2. If you plan to attend in person, the organizing committee will provide accommodation details separately.
3. Please prepare your final camera-ready manuscript according to the formatting guidelines available at https://mathai.club and upload it to OpenReview by March 15, 2026 18:00 Moscow time.

Should you have any questions regarding the program, logistics, or your presentation slot, please do not hesitate to contact us.

We look forward to your contribution to MathAI 2026.

With kind regards,

MathAI 2026 Program Committee
International Conference on Mathematics of Artificial Intelligence
https://mathai.club
OpenReview: https://openreview.net/group?id=mathai.club/MathAI/2026/Conference
Telegram: https://t.me/MathAI_club
Email: mathai.club@yandex.ru